# Non-Antibiotic Prophylaxis for Recurrent UTIs in Neurogenic Lower Urinary Tract Dysfunction (NAPRUN): Study Protocol for a Prospective, Longitudinal Multi-Arm Observational Study

**DOI:** 10.3390/mps6030052

**Published:** 2023-05-24

**Authors:** Fabian P. Stangl, Laila Schneidewind, Bernhard Kiss, Jennifer Kranz, Florian M. Wagenlehner, Truls E. Bjerklund Johansen, Béla Köves, Jose Medina-Polo, Ana Maria Tapia, Zafer Tandogdu

**Affiliations:** 1Department of Urology, University Hospital of Bern, 3010 Bern, Switzerland; 2Department of Urology, University Medical Center Rostock, 18055 Rostock, Germany; 3Department of Urology and Paediatric Urology, Uniklinik RWTH Aachen, 52074 Aachen, Germany; 4Department of Urology and Kidney Transplantation, Martin Luther University, 06120 Halle (Saale), Germany; 5Clinic for Urology, Pediatric Urology and Andrology, Justus-Liebig-University Giessen, 35390 Giessen, Germany; 6Institute of Clinical Medicine, University of Aarhus, 8200 Aarhus, Denmark; 7Department of Urology, Oslo University Hospital, 0315 Oslo, Norway; 8Institute of Clinical Medicine, University of Oslo, 0315 Oslo, Norway; 9Department of Urology, University of Szeged, 6725 Szeged, Hungary; 10Department of Urology, Hospital Universitario 12 de Octubre, 28041 Madrid, Spain; 11Department of Urology, Hospital Universitario Río Hortega, 47012 Valladolid, Spain; 12Department of Urology, University College London Hospitals, London W1G 8PH, UK

**Keywords:** non-antibiotic prophylaxis, urinary tract infection, urinary tract dysfunction, NLUTD, self-catheterization, Angocin, UroVaxom, StroVac, bladder irrigation, D-mannose

## Abstract

Introduction: Patients with neurogenic lower urinary tract dysfunction (NLUTD) reliant on intermittent self-catheterization for bladder emptying are at an increased risk of recurrent urinary tract infections (rUTI). So far, the most common practice in the prevention of rUTIs is long-term low-dose antibiotic prophylaxis, phytotherapy, and immunomodulation, whereby antibiotic prophylaxis inevitably leads to the emergence of drug-resistant pathogens and difficulty in treating infections. Therefore, non-antibiotic alternatives in the prevention of rUTIs are urgently required. We aim to identify the comparative clinical effectiveness of a non-antibiotic prophylaxis regimen in the prevention of recurrent urinary tract infections in patients with neurogenic bladder dysfunction who practice intermittent self-catheterization. Methods and analysis: In this multi-centre, prospective longitudinal multi-arm observational study, a total of 785 patients practising intermittent self-catheterisation due to NLUTD will be included. After inclusion, non-antibiotic prophylaxis regimens will be instilled with either UroVaxom^®^ (OM-89) standard regimen, StroVac^®^ (bacterial lysate vaccine) standard regimen, Angocin^®^, D-mannose (oral dose 2 g), bladder irrigation with saline (once per day). The management protocols will be pre-defined, but the selection of the protocol will be at the clinicians’ discretion. Patients will be followed for 12 months from the onset of the prophylaxis protocol. The primary outcome is to identify the incidence of breakthrough infections. The secondary outcomes are adverse events associated with the prophylaxis regimens and the severity of breakthrough infections. Other outcomes include the exploration of change in susceptibility pattern via the optional rectal and perineal swab, as well as health-related quality of life over time (HRQoL), which will be measured in a random subgroup of 30 patients. Ethics and dissemination: Ethical approval for this study has been granted by the ethical review board of the University Medical Centre Rostock (A 2021-0238 from 28 October 2021). The results will be published in a peer-reviewed journal and presented at relevant meetings. Study registration number: German Clinical Trials Register: Number DRKS00029142.

## 1. Introduction

### 1.1. Neurogenic Lower Urinary Tract Dysfunction (NLUTD)

An increased risk of rUTIs is known in patients with neurogenic lower urinary tract dysfunction (NLUTD) due to the impairment in the urinary tract’s physical protective mechanisms. The cause of NLUTD and how it can manifest as a UTI are heterogeneous. Patients with NLUTD have an annual incidence of 54.9% of developing UTIs [1]. Unfortunately, most patients with NLUTD who develop UTIs will have recurrent episodes of infections followed [2]. The UTIs in this population are not only a worry for mortality and morbidity but also have a great impact on quality of life. Moreover, frequent UTIs mean that these patients get exposed to a high quantity of antibiotics, which is the main driver of the emergence of drug-resistant difficult-to-treat bacterial infections.

To break the vicious cycle of recurrent UTIs (rUTI), prophylactic measures are propagated, albeit with limited evidence to support one intervention over the other. So far, the most common practice in the prevention of rUTIs is long-term low-dose antibiotic prophylaxis, phytotherapy, and immunomodulation. The antibiotic prophylaxis option exacerbates the emergence of difficult-to-treat drug-resistant infections. Therefore, non-antibiotic alternatives in the prevention of rUTIs are urgently required. 

We have recourse to a broad armamentarium of non-antibiotic options, whereby significant evidence is limited. Several viable and readily available options have been shown to yield consistent results. Main alternatives to antibiotic prophylaxis include vaccines, antiseptics, urine-specific antimicrobials, and direct mechanic washouts of the bladder. A variety of vaccines have been used in the prevention of rUTIs. Especially OM-89 (Urovaxom^®^) and the bacterial lysate vaccine Strovac^®^ have been diligently tested and proven to lower the rate of UTI [3,4,5,6]. Additionally, a wide variety of phytotherapeutics is currently in use, especially in Eastern medicine, as a cost-effective and low-threshold, whereas ambivalent efficacy has been reported. A combination of Tropaeoli majoris herba (Nasturtium) and Armoraciae rusticanae radix (Horseradish) demonstrated excellent safety and efficacy in the prevention of recurrent UTIs [7,8]. D-Mannose is another beneficial and cheap prophylactic, therapeutic agent in rUTI with very few side effects [9,10]. Bladder irrigation with various agents, e.g., neomycin, acetic acid, or sterile saline, is also widely used in order to lower bacterial load and defer sedimentation [11,12,13]. 

The main aim of our study is to identify the comparative clinical effectiveness of non-antibiotic prophylaxis regimens in the prevention of rUTI infections in patients with neurogenic bladder dysfunction who practise clean intermittent self-catheterization (CISC). The primary outcome is depicted by the number of breakthrough infections. Secondarily adverse events associated with the prophylaxis regimens and the severity of breakthrough infections must be investigated. 

Change in susceptibility pattern via the optional rectal and perineal swab, as well as health-related quality of life over time (HRQoL) outcomes, will be measured in a random subgroup of 60 patients.

The primary objective is to identify the clinical effectiveness of different prophylactic regimens. The secondary objective is to determine the adverse events related to prophylaxis and the severity of breakthrough infections. 

### 1.2. rUTI in NLUTD

Patients unable to empty their bladder to completion due to a neurogenic disorder require adjunct management options to ensure a low-pressure urinary tract as well as sufficient emptying of the bladder. The most common way of achieving bladder emptying in patients with manual dexterity or with a carer to support is CISC. In this population, the rate of rUTI is significantly high at 9.4 episodes per year [14,15]. 

### 1.3. Definition of UTI in NLUTD

In our study, UTIs in this challenging collection of patients are defined as follows:○Positive urine culture with typical bacteria in UTI with >10^2^ CFU/mL (Colony forming units);○Urinary urgency;○Urinary frequency;○Dysuria;○Fever/chills;○Worsening of a neurological condition;○Altered mental status;○De novo incontinence.

Diagnosis of UTI is fulfilled if patients suffer from two or more of the above symptoms with positive urine culture (mandatory) [16]. 

### 1.4. Bacterial Spectrum in rUTI in NLUTD

There are very few publications on the specific bacterial spectrum in rUTI in NLUTD. Mainly the bacterial spectrum of complicated and nosocomial acquired UTI is taken as representative of rUTI as well [17].

Recent data show a similar bacterial spectrum to other populations suffering from rUTI. Most frequently, *Escherichia coli*, *Enterococcus faecalis*, *Klebsiella pneumoniae/variicola*, *Streptococcus viridans*, *Pseudomonas aeruginosa*, and coagulase-negative Staphylococci have been detected. To date, unfortunately, fluoroquinolones are the most frequently used antibiotic in the management of UTI, whereby no considerable increase in resistance could be observed. Identified *E. coli* strains showed a noteworthy resistance increase against amoxicillin/clavulanic acid from 26 to over 38% [18].

### 1.5. Management of rUTI in NLUTD

Treatment regimens in handling rUTI are highly individual and often challenging. Due to a lack of clinical information, grading of severity and, therefore, indication for antibiotic treatment is often unclear. Even though we usually have specific information concerning causative organisms for individual patients, empiric treatment regularly fails the patient. A multitude of antibiotic and non-antibiotic prophylaxis regimens have been tested to date, yet missing conclusive evidence [14,19,20,21].

### 1.6. Rationale

To our knowledge, there is limited evidence on prophylaxis to prevent rUTIs in NLUTD patients practising CISC. 

Long-term health outcomes and quality of life are significantly impaired in our population of interest, and increasing antibiotic consumption, as well as rising resistance rates against our antibiotic armamentarium, necessitates a controlled evaluation of non-antibiotic alternatives in the reduction of UTI frequency. Few trials support the use of non-antibiotic prophylaxis, but further investigation in a prospective manner is demanded eventually. If solid data supports any kind of named non-antibiotic alternative, this could serve to lower the burden on our healthcare system financially on the hand, and on the other hand, the even more pressing question of how to offer an evidence-based substitute in prophylaxis and therefore lowering exposure in the worldwide development of antibiotic resistance.

## 2. Objectives

The aim of this prospective observational cohort study is to investigate the impact of five different non-antibiotic prophylaxis regimens over the course of one year on predefined time points. Our long-term goal is to identify safe and effective candidate regimens that could be carried to a randomised controlled trial (RCT) to establish clinical and cost-effectiveness. 

### 2.1. Primary Objectives

To identify the clinical effectiveness of non-antibiotic prophylaxis regimens in the prevention of episodes of UTIs in NLUTD patients managing their bladder with CISC and having rUTIs. 

### 2.2. Secondary Objectives

○Number of UTIs under non-antibiotic prophylaxis necessitating antibiotic treatment;○Evaluation of antibiotic resistance of inherent microbial flora and acquired urine cultures;○Evaluation of antibiotic consumption (DDD-defined daily doses) under non-antibiotic prophylaxis;○Evaluation of quality of life in patients undergoing non-antibiotic prophylaxis.

## 3. Design

This is a prospective, longitudinal, multi-arm, multi-centre, observational, clinical cohort study. Patients will receive approved non-antibiotic UTI prophylaxis with either *Escherichia coli* viva OM-89 (UroVaxom^®^), bacterial lysate vaccine (StroVac^®^), nasturtium and horseradish root powder (Angocin^®^), D-Mannose or bladder irrigation with saline once daily. Occurring complications under named prophylaxis will be treated according to the current standard of care. 

Primary Outcome Measure: Incidence of symptomatic breakthrough infections.

Secondary Outcomes: Severity of infectious breakouts with antibiotic use, development of antibiotic-resistant (AMR) flora, number of antibiotics used, quality of life, and exploration of health economic assessment variables. 

## 4. Participants

Adult patients with neurogenic bladder dysfunction, practising clean intermittent self-catheterization, who suffer from recurrent UTI (used definition: more than 3 episodes in the last 12 months) and who failed conservative measures. Non-antibiotic prophylaxis (UroVaxom, StroVac, Angocin, D-mannose, saline bladder irrigation) will be offered to these patients per-protocol basis. The data of the bacterial isolates and their susceptibility results used in this study will be obtained as part of routine clinical care. The necessity of informed patient consent is at the discretion of the participating institution. All patient data will be analysed anonymously.

### 4.1. Inclusion Criteria

Microbiologically proven UTI according to Section 1.3;Age > 18;Ability to consent;NLUTD with practised ISC;rUTI (used definition: more than 3 infections per year) independent from a bacterial spectrum of infection;Failed conservative measures (e.g., an increase in fluid intake and an increase in assisted bladder emptying);Previous episodes of UTIs do not show multi-drug-resistant bacteria;Non-antibiotic prophylaxis with UroVaxom, StroVac, Angocin, D-Mannose or bladder irrigation with saline;Signed informed consent form.

### 4.2. Exclusion Criteria

Patients practising ISC with augmented bladder or continent pouches;Inability to consent;Intention-to-treat analysis for dropouts;Failure to provide consent;MDR pathogens in previous episodes of UTIs;Antibiotic prophylaxis in the months before study enrolment.

### 4.3. Consent

Data collection in centres in Germany and Switzerland will fall under general consent. If necessary, because of country-specific ethical board requirements, patients will need to give written consent. Consent will be obtained by the primary physician. 

## 5. Patients Recruitment

Identification of observations: Patients to be observed will be identified through participating clinicians in either an outpatient setting, hospitalised patients or follow-up visits after symptomatic infections. The eligibility criteria checklist of all cases considered to be observed will be filled in by participating principal investigators (PI). Clinical information from the initial diagnosis should be used to assess eligibility.

A logbook of patient screening will be kept to keep track of the number of patients per unit who are eligible and who are not. We will also keep track of those who are eligible and refuse to participate. If recruitment rates are slower than the anticipated trajectory, the logbook will be reviewed to identify obstacles in recruitment. 

Patients will be stratified to balance our groups according to age, sex, underlying neurological condition and number of infections prior to study inclusion. 

## 6. Observations

### 6.1. Demographic Characterization—Baseline

Age;Country;Sex;BMI;Underlying neurological disease;Date of initial diagnosis;Charlson Comorbidity Index;Medication;ACSS score;Practice of sterile intermittent self-catheterization;Practice of clean intermittent self-catheterization;Duration of practised catheterisation in months;Number of catheterisations per day;Number of UTIs in last 12 months;Hospitalisation due to UTI;Earlier prophylaxis regimens/sufficiency of prophylaxis (prophylaxis regimen with duration and dosing, Patient impression, reduction of UTI in three months);Appropriateness of intermittent self-catheterization (measured by post-void residual (PVR) via ultrasound, which should be less than 100–150 mls).

### 6.2. Non-Antibiotic Prophylaxis Regimen

UroVaxom (OM-89) will be administered as a standard regimen: once daily for three months with a three-month pause following and three ten-day cycles at 7, 8 and 9 months.,StroVac (bacterial lysate vaccine) standard regimen: three vaccines in 6–12 weeks with two weeks between each vaccination *.Angocin twice daily for the duration of the study.D-mannose (oral dose 2 g) once daily for the duration of the studyBladder irrigation with saline (0.9%; 250 mL) once per day for the duration of the study.

* In case of restricted drug availability due to country-specific authorisation of vaccines, additional treatment arms with similar mechanisms of action will be opened and analysed as a subgroup. If necessary, amendments to the protocol will be issued, and additional ethics approval will be requested.

### 6.3. Evaluation at 3, 6, 9 and 12 Months (Table 1)


Number of UTI (reported by patient; if externally diagnosed and treated by, e.g., primary caregiver records will be obtained and added to study documentation);Hospital admission due to UTI or severe systemic infection (SOFA Score, SIRS criteria, Antibiotic administration, duration of hospital stays, urine culture);Antibiotic treatment/ treatment with a reserve antibiotic;Urine culture, including susceptibility profile;Rectal swab (optional);Perineal swab (optional);ACSS score;Adverse events during intake;HRQoL measures (subgroup of 30 patients);End at 12 months: patient impression (Likert Scale).


**Table 1 mps-06-00052-t001:** Summary of data collection points.

	Day 0	Month 3	6	9	12
*ACSS*	*X*	*X*	*X*	*X*	*X*
*Recruit*	*X*				
*Rectal swab*	*X ^*		*X ^*		*X ^*
*Perineal swab*	*X ^*		*X ^*		*X ^*
*CRF*	*X*	*X*	*X*	*X*	*X*
*Urine culture*	*X*		*X ^*		*X*

^ optional; ACSS = Acute Cystitis Symptom Score; CRF = Case Report Form.

### 6.4. Documentation of Stop Criteria/Study Dropout

Reason for dropout;Time Point of dropout;UTI;Sepsis;Adverse events;Problems of administration of the drug;Duration of non-antibiotic regimen;Switch to another prophylaxis regimen/which one.

### 6.5. Blinding of Study Assessors

Study assessors will not be blinded.

## 7. Statistical Considerations

Power analysis was performed to determine an adequate sample size. We considered that these patients suffer 9.4 UTIs per year with an SD of 5.414. Furthermore, we assumed that a reduction of 3 UTIs per year is clinically significant. In order to achieve an effect size of 0.1, 785 participants are necessary (power = 0.87; alpha = 0.05). Furthermore, to achieve an effect size of 0.2, 197 participants are necessary, or for an effect size of 0.3, 88 participants are required, respectively. Data measured on a continuous scale will be expressed as mean, standard deviation, range, and median. Counts and percentages of patients in the categories will be used to convey categorical data. When necessary, T- or Mann-Whitney tests will be performed to determine statistical differences in continuous variables, and Chi-square or Fisher’s tests will be used to test for statistical differences in categorical variables.

Patients with a minimum follow-up of 6 months who remained on the same prophylaxis regimen will be eligible for the final statistical analysis. 

### 7.1. Patient and Public Involvement

Patients or the public were not involved in the design, conduct, reporting, or dissemination plans of our research.

### 7.2. Protocol Amendments

If protocol amendments are necessary, all involved parties (e.g., participating and recruiting institutions, local investigators, data analysts etc.) will be notified before relevant changes come into place.

### 7.3. Trial Audit

The trial will be audited by primary investigators after each data collection point. If any adverse events occur in between, those will be investigated, addressed, and reported accordingly.

## 8. Management

The Global Prevalence Study on Infection (GPIU) study team and platform will provide adequate data management.

## 9. Administration, Logistics & Quality Assurance

### 9.1. Sharing of Observations

The collection of observational data will be performed using electronic (e) case report forms (CRF). Sites will receive direction from the study coordinator to help them complete the eCRFs. The research team retains the right to modify or enhance the eCRF template as necessary. Sites should utilise revised or extra forms, as these changes do not constitute a protocol amendment.

### 9.2. Central Data Monitoring

The study coordinator will examine the completed eCRFs for consistency with the protocol, protocol compliance and missing variables. If any data gaps or anomalies are discovered, the site will raise questions for resolution. It is possible to query any systemic discrepancies found through central data monitoring.

### 9.3. Definition of End of Study

The end of the study will be the date of the last observation captured of our last enrolled participant.

## 10. Research Governance

### 10.1. Sponsor Responsibilities

To date, no sponsors have been named.

### 10.2. Participating Sites Responsibilities

Responsibilities delegated to participating sites are defined in an agreement between the Sponsor and the individual site.

## 11. Participant Protection and Ethical Considerations

### 11.1. Approvals

The cohort study is designed longitudinal, observational with no intervention with no patient identifiable data and, generally, should not need ethics review. As this might differ between countries, the investigating sites should retrieve ethical consent according to their local needs. Ethical approval was granted in Germany by the ethical review board of the University Medical Centre Rostock (A 2021-0238 from 28 October 2021). In case of a change of protocol, amendments from country-specific ethical committees will be requested. 

### 11.2. Conduct

This registry will be conducted according to the approved protocol and its amendments, supplementary guidance, and manuals supplied by potential sponsors.

### 11.3. Data Protection Act (DPA)

Data collected during the course of the study will be kept strictly confidential and accessed only by members of the team. Participants’ details will be stored on a secure database under the guidelines of the 1998 Data Protection Act, and regular checks and monitoring are in place to ensure compliance. Data are stored securely in accordance with the Act and archived in a secure data storage facility. The IT manager at Rostock University, Germany (in collaboration with the Study Directors and study coordinator) will manage access rights to the data set. Participants will be allocated an individual-specific number, and their details will be anonymised on the secure database. We anticipate that anonymised data may be shared with other researchers to enable international prospective meta-analyses. To comply with the 5th Principle of the Data Protection Act 1998, personal data will not be kept for longer than is required for the purpose it has been acquired.

## 12. Publication Policy

The main results will be published in peer-reviewed journals on behalf of all collaborators. The manuscript will be prepared by a writing group of members of the scientific committee and the European Section of Infections in Urology. Group authorship will be inserted under the collective title of ‘the NAPRUN Study Management Group’. If one or more individuals have made a significant contribution above and beyond other group members, but where all group members fulfil authorship rules, authorship will be attributed to the named individual(s) and the NAPRUN Group. Participating clinicians may be selected to join the writing group based on intellectual and time input. All participating clinicians will be acknowledged in the publications. Any presentations and publications relating to the data must be authorised by the GPIU group.

## Data Availability

Data will be available on request due to restrictions e.g., privacy or ethical. The acquired data in this study will be available on request from the corresponding author. The data won’t be publicly available due to the General Data Protection Regulation by the European Union.

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
