# Peer review of "Non-Antibiotic Prophylaxis for Recurrent UTIs in Neurogenic Lower Urinary Tract Dysfunction (NAPRUN): Study Protocol for a Prospective, Longitudinal Multi-Arm Observational Study"

_mps, 2023, doi:10.3390/mps6030052_

Round 1

Reviewer 1 Report

- Scientific Relevance: The topic addressed is adequate to the journal, may have utility in directing health policies, and has a fair amount of scientific relevance; however, it does not contain any results and does not provide data from which conclusions can be drawn about the effectiveness of the proposed solutions. Based on these shortcomings, the paper does not contribute significantly to increasing scientific knowledge on the subject.

- The sections: materials and methods, results, discussion and conclusions are not present. Objectives are repeated several times in a confusing manner within the text (at the end of Introduction 97-108 and in section 2. Objectives). The paper is not formatted according to the journal guidelines. Overall, it isn't considerable as a methodologically well-developed paper.

- General organization of the paper: The number of paragraphs is excessive and many of them consist of only a few lines. Table 1 (321) has title and comments on the same line. In the same table, the CRF entry is indicated with lowercase x instead of uppercase X.

- Bibliography: The bibliography is up to date but there are no bibliographical references in the paper beyond the introduction. No obvious self-citations are present.

- Tips for the authors: try a new submission by setting the paper differently and reporting results that demonstrate the efficacy of the proposed non-antibiotic therapies.

Minor editing of English language required

Author Response

Dear Reviewer III

Thank you very much for your comments and your thorough consideration. Please let us address your concerns and elaborate.

Scientific Relevance: We think we clearly stated the knowledge gap present regarding non-antibiotic prophylaxis in NLUTD and therefore feel there is an absolute need for robust and reliable data. The sole purpose of this study to evaluate if there is an effect with any of named therapies. If we identify a significant reproducible effect, we will use this knowledge to design and conduct an RCT testing our hypothesis against the current standard of practice namely antibiotic therapy. Our working group is heavily involved in infectious diseases in urology, and we try to identify different approaches to reduce antibiotic consumption. The number of rUTI in our study population is extremely high and antibiotics are regularly used subsequently. This offers a huge potential for reducing antibiotic use.

Sections: We changed our protocol after first submission suggested by the editors and got positive feedback regarding the accordance to journal guidelines.

General Organization: We edited the table according to your suggestions.

We hope our response supports reconsideration of our protocol. We are eager to conduct scientifically sound research and are therefore grateful for your critical comments, which surely help improving our planned study.

Reviewer 2 Report

I am not certain that the initial sample size of 785 will be enough for five different experimental groups (N = cca 150 / group minus drop-out). Furthermore, there is a strong possibility for a non-symmetrical distribution in experimental groups, as pointed out in line 46: "The management protocols will be pre-defined, but selection of protocol will be at the clinicians' discretion." Please address these concerns. 

I don't see the point of the secondary outcome, health-related quality of life over time (HRQoL) -as it will be measured in a drastically small number of patients (N=30, as in 6 /group???). Please omit this parameter, or increase the sampling size.

The manuscript is referring to a Figure that is missing, as pointed out in line 203: "Occurring complications under named prophylaxis will be treated according to the current standard of care (Figure 1)." Please provide the Figure.

Author Response

Dear Reviewer 1

Thank you very much for your valuable comments. We performed a thorough power analysis using data from current literature. Since the absolute number of UTI per year in the study population is very high, sample size needs to relatively small in order to achieve a significant effect size. We hope this dispels your concerns. Regarding the non-symmetrical distribution, we share your thoughts and tried to even this bias by stratified recruitment. This means that we will the decision to the primary physician but will inform participating centers if recruitment is skewed. We would like to facilitate organic growth of groups without aggressive interference but will aim to achieve five balanced groups with evenly matched patients (elaborated under 5. Patient recruitment).

Regarding the HRQoL subanalysis we increased sample size to 60 patients. We do not strive to assess the QoL per group but rather want to gain an overall overview of QoL in our study population to add value to our planned RCT and perform a robust power analysis for further QoL analysis.

We deleted the reference.

We would like to express our gratitude for your kind comments and considerations and think your review helped improving our protocol substantially.

Reviewer 3 Report

The issue of improving treatment/prophylaxis of rUTI in patients with NLUTD is very interesting and prudent, and prospective comparative studies highly needed, especially such, that can find a way forward without antibiotics.

I have some comments for the authors to consider:

1. While the aim of the study apparently is to find treatments, that can be incorporated in a randomised study, i wonder whether the present setup will be helpful; the patients are included as based on the discretion of the clinicians involved but this may heavily bias the outcome of the study, and i am not sure, whether a future randomised study can be based on such a design ? The choice of treatmnet will depend on the experience of the clinicians and they would probably not be prone to start a treatment let us say with a vaccine, if they think bladder irrigation is the best. It would be optimal to perform this study with some kind of randomisation, perhaps with fewer arms ? One problem could be, that the effect of vaccines may take some time to develop being dependent on immunti needing some time to build up. To compare with e.g. Mannose, which starts acting right away, may reduce rUTI  from the beginning, while the effect of a vaccine on rUTI may need months. The design should then take this into consideration e..g postponing the start of the Mannose treatment. Another aspect of e.g. UROVAC is - what i cannot find data for in the literature - that as based on E.coli components mostly work on E.coli and not so much on e.g. enterococci. And the prevalence of E.coli in this type of patients may then be too low to show effect on rUTI as compared with "healthy" women with rUTI. One way of circumventing this problem could be to study the patient as his own control i.e. an observation period of 6 months and then one of the treatments, comparing the next 6 months with the patients´own history.

2. UTI occurring after start of treatment, the main outcome measure -  needs a strict definition, perhaps more strict the the inclusion definition. The criteria of 10-2 CFU/ml demands meticulous microbiology i.e. sampling of higher volumes of urine, preferably 0.5 ml.

3. I am curious why there is not an arm for the standard treatment as mentioned in the introduction, namely antibiotic prophylaxis ? The other treatment arms are selected to be non-antibiotic, but then you need this for comparison.

4. Some minor points:

Line 71: Explain QoL - it is later used written out.

Line 73: "Drug resistant.." what?

Line 98: There is a "so", which does not give meaning

Line 143: "no considerable increase" must be the opposite, e.g. "considerable increase" ? We are talking about fluorquinolones which are prone to select for resistance.

Line 169: Better to use "exposusre" than "pressure"

Line 233: Is this an inclusion criterium ?

Author Response

Dear Reviewer 2

Thank you very much for your valuable comments.

We would like to express our gratitude for your kind comments and considerations and think your review helped improving our protocol substantially. Please let us address your major points first:

We are aware of the complex task at hand of finding suitable treatment options. Planning a randomized controlled trial with potentially significant results is extremely difficult, given the current literature. If we do set up a trial, we would like to have some sort of guidance regarding the choice of therapeutics. The treatment options we would like to investigate in this trial are diverse and depict maximally different approaches. Regarding the non-symmetrical distribution, we share your thoughts and tried to even this bias by stratified recruitment. This means that we will the decision to the primary physician but will inform participating centers if recruitment is skewed. We would like to facilitate organic growth of groups without aggressive interference but will aim to achieve five balanced groups with evenly matched patients (elaborated under 5. Patient recruitment). Given your interesting thought of a case-control-study we feel that the only reliable variable in this population is infection frequency, albeit not the absolute number just the increased frequency. Therefore, we are confident to identify significant changes in infection recurrence. A subsequent RCT would be set up according to standard guidelines with blinding to reduce bias. We discussed our design extensively with methodologist and other experts in the field and concluded that in order to gain the desired knowledge in a timely fashion we have to go off the given path and use an innovative design. But the results this study will hopefully yield cannot be used to influence patient counselling and are solely for the purpose of impactful further research.

Regarding the definition of UTI, we narrowed it down to very concrete points (1.3. definition of UTI) but are very aware of the complexity of this task. There is no available uniform definition in this patient cohort and therefore we set cornerstones combining different guidelines. We will only include centers who can meet our quality standards and multiple studies have shown that bacterial growth of 105 CFU is too high of a cut off, potentially missing ongoing infection.

We are just comparing non-antibiotic options against each other to investigate if any yields significant results. If we meet our defined level of significance, we will compare the treatment/therapeutic against antibiotic prophylaxis.

Lastly, we would like to address your minor points:

We corrected the mistakes in line 71, 73, 98 and 169. Regarding line 143 we highly agree, but only state the most recent literature in this patient cohort. In line 233 we state that patients must be already assigned to non-antibiotic prophylaxis which helps our study design and its planned multinationalism, since we will conduct a non-interventional and solely observative study.

We would like to express our gratitude for your kind comments and considerations and think your review helped improving our protocol substantially.

Round 2

Reviewer 1 Report

The paper is well written, proposes a well-detailed methodology and addresses a topic of great importance in the field of infectious disease treatment. No further changes need to be made and it can be published in its current form.

Reviewer 2 Report

Dear Editors,

I am satisfied with the author's response to my remarks. I think that the manuscript can be accepted, as it has been modified adequately. 

Kind regards,

Darko Modun

Reviewer 3 Report

The authors responded adequately to comments from reviewer